# Can Vaginal Seeding at Birth Improve Health Outcomes of Cesarean Section-Delivered Infants? A Scoping Review

**DOI:** 10.3390/microorganisms13061236

**Published:** 2025-05-28

**Authors:** Phoebe LaPoint, Keona Banks, Mickayla Bacorn, Ruhika Prasad, Hector N. Romero-Soto, Sivaranjani Namasivayam, Qing Chen, Anal Patel, Shira Levy, Suchitra K. Hourigan

**Affiliations:** Clinical Microbiome Unit, National Institute of Allergy and Infectious Diseases, National Institutes of Health, Bethesda, MD 20892, USA; phoebe.lapoint@nih.gov (P.L.); keona.banks@nih.gov (K.B.); mickayla.bacorn@nih.gov (M.B.); ru.prasad@nih.gov (R.P.); hector.romerosoto@nih.gov (H.N.R.-S.); sivaranjani.namasivayam@nih.gov (S.N.); qing.chen@nih.gov (Q.C.); anal.patel@nih.gov (A.P.); shira.levy@nih.gov (S.L.)

**Keywords:** microbiome, neonate, delivery mode, maternal–child seeding

## Abstract

Although Cesarean section (C-section) delivery is often a necessary medical intervention, it also increases the risk of the infant developing chronic inflammatory, metabolic, and neurodevelopmental disorders. The association of C-section with the development of these conditions is thought to be partially mediated by the effects of the C-section on the infant’s microbiome development and subsequent immune regulation. C-section-delivered infants acquire a different set of microbes compared with infants who are vaginally delivered. “Vaginal seeding” exposes C-section-delivered infants to the maternal vaginal microbiome directly after birth, partly replicating the microbial exposures they would have received during a vaginal delivery. Studies have shown that vaginal seeding at birth partially restores the infant microbiome towards that of a vaginally delivered infant. More recently, preliminary studies have shown a potential benefit of vaginal seeding on health outcomes. Here, we examine the evidence from observational studies and randomized controlled trials that have evaluated microbiome restoration after C-section, and we discuss new research assessing the potential impact of vaginal seeding on immune, metabolic, and neurodevelopmental outcomes and the underlying mechanisms. Collectively, we review the potential health benefits, safety risks, regulatory implications, and future directions for the use of vaginal seeding in infants delivered by C-section.

## 1. Background

The microbes that first colonize an infant’s body at birth play a critical role in immune system education and metabolic programming [1]. These pioneering colonizers are opportunistic microbes found within the infant’s immediate surroundings during birth [2]. It is estimated that maternal microbial communities contribute approximately 59% of this initial infant microbiome colonization [3]. This initial colonization affects the order and timing of subsequent colonization by microbial species as they begin to fill specific niches [2,3,4]. Disruption to this early life microbial colonization of an infant can have lasting metabolic, immune, and inflammatory consequences [5].

Cesarean section (C-section) delivery represents a major disruptor of early life microbial colonization [2,6]. During birth, vaginally delivered infants are exposed to the maternal vaginal and perineal microbiome, and these microbes are the infants’ initial colonizers [2,3]. In contrast, C-section-delivered infants bypass this initial exposure and are mainly colonized with skin and environmental bacterial communities, which can result in suboptimal development of the infant’s microbiome [2,3,7,8]. C-section-delivered infants are also at increased risk of inflammatory, metabolic, and neurodevelopmental disorders later in life, which are believed to be partially influenced by the microbiome [9].

“Vaginal seeding”, or vaginal microbial transfer, is a type of maternal microbial seeding which exposes a C-section-delivered infant to their own mother’s vaginal microbiome, which they would typically be exposed to during a vaginal delivery [10]. This process involves first inserting a sterile gauze into the mother’s vagina prior to delivery and then wiping the gauze over the infant following the C-section [10]. Vaginal seeding is hypothesized to partially restore the microbiome disrupted by C-section delivery, potentially reducing the risk of developing conditions that are at greater risk following a C-section. Given the growing interest in vaginal seeding from pregnant mothers and healthcare professionals, it is essential to assess whether this process is efficacious and safe [11]. In this review, we highlight the rationale for maternal microbial seeding of C-section-delivered infants, as well as evidence of its efficacy to change the infant microbiome and improve health outcomes, challenges, safety concerns, regulatory and ethical issues, and future opportunities to advance this research.

## 2. Importance of Infant Microbiome Development in Immune and Metabolic Education and Regulation

The gut microbiome is known to heavily impact host immune function and metabolism [12,13,14,15]. During the first few years of life, there is dynamic development of the infant gut microbiome [16]. Studies indicate that in this critical postnatal period, the microbial environment has the potential to profoundly influence the education and development of both the immune system and metabolic pathways [1,5,17,18]. In the first months, there is a rapid increase in bacterial diversity, with the first few days being dominated by facultative aerobic Enterobacteriaceae before changing to mainly strict anaerobes with high abundances of *Bifidobacterium* and *Bacteroides* [6]. After the cessation of breastfeeding and the introduction of solid food, the gut microbiome composition starts to change to a more adult-like structure [6,7,10]. This suggests that an infant’s early life events, including the process of birth that facilitates initial microbial colonization, may greatly impact the development of the immune system and metabolic pathways, leading to subsequent effects on the infant’s long-term health outcomes.

### 2.1. The Impact of the Developing Gut Microbiome on Immune System Education

The developing gut microbiome and gut microbial products regulate and educate immune maturation, which sets the stage for life-long host–microbial interactions and immune homeostasis [1,19]. This period, beginning prenatally and lasting until approximately 3 years of age, is referred to as the “window of opportunity”, as many exposures such as birth mode, diet, and antibiotic use can cause aberrant host–microbial interactions, potentially leading to inflammatory and metabolic conditions later in life. As early as in pregnancy, fetal immune development is supported by microbial metabolites originating primarily from the maternal microbiota [20]. At birth, the emerging immune system of the infant has its first substantial microbial exposure. During this period, neonatal gut innate immune cells are first responders and, together with the gastrointestinal epithelial barrier, protect against invading pathogens and promote beneficial interplay with the commensal microbiota that is being established [21,22]. Appropriate regulation of these responses is essential to prevent damaging inflammation at mucosal sites during the establishment of the gut microbiome in early life, as well as to generate protective responses against pathogenic microbes. Breast milk supports this process by providing passive immunization and human milk oligosaccharides that promote growth of beneficial bacterial and by directly stimulating the neonatal immune system [23,24]. While a component of the adaptive immune system is present prenatally in the fetus, there is rapid expansion postnatally, which accelerates following the introduction of solid food, with a large shift in the gut microbiome to a more adult-like profile [6,7,10,19].

The developing gut microbiome is crucial in regulating the local immune system. The very first microbial colonizers during and after birth promote tumor necrosis factor (TNF) secretion by monocytes and macrophages [18]. This microbiota-derived TNF is essential in inducing the differentiation and functional maturation of certain subsets of dendritic cells, which are responsible for capturing and presenting antigens to T cells [18]. The interplay between the gut microbiota and intestinal epithelial wall is also critical in controlling the local mucosal immune system. Commensal microbes interact with receptors on the intestinal epithelial wall, which in turn can secrete anti-microbial peptides (AMPs) which help fight pathogens and are anti-inflammatory [25,26]. Moreover, gut microbial products, especially short-chain fatty acids (SCFAs), have been shown to have a protective role. SCFAs aid in the production of AMPs, which help and stimulate the maturation and expansion of colonic regulatory T cells, which have an anti-inflammatory effect [25,26]. SCFAs also contribute to the modulation of the intestinal epithelial barrier and induce the proliferation of innate lymphoid cells [26]. Furthermore, there is increasing recognition that the gut microbiota can regulate systemic immune responses. A well-characterized example of this is the gut microbiome affecting the differentiation of systemic T cell populations into T-helper (Th) Th1, Th2, and Th17 cells or into T cells with a regulatory phenotype, partly via SCFA production [27].

The importance of early life gut microbiome colonization on immune regulation is well highlighted in germ-free (GF) rodents that lack any microbes, resulting in major defects within the spleen, thymus, and lymph nodes due to improper lymphoid tissue development [28,29]. Importantly, GF mice that are colonized later in life show a different transcriptional profile in the jejunum and colon, compared to that of conventionally raised specific-pathogen-free mice, and fail to develop certain immune cell types [17,30]. This suggests that in addition to the presence of microbes, the timing of microbial colonization is important in intestinal immune development.

Early life exposures during the critical window of opportunity have the potential to disrupt gut microbiome colonization, maturation, and immune regulation. In addition to C-section delivery, which will be discussed in detail below, exposures such as antibiotics and environmental factors during this period can affect microbiome-regulated immune system education, leading to susceptibility to immune and inflammatory diseases. Examples from epidemiological studies include increased risk of inflammatory diseases including asthma, atopy, and inflammatory bowel disease later in life due to early life antibiotic exposure, with murine models indicating a microbiome-mediation-based cause [31,32,33,34,35,36,37]. Conversely, environmental factors such as living on a farm and having siblings in the home can advance gut maturation and decrease the risk of atopic disease [38,39].

### 2.2. Microbiome Impact on Metabolic Regulation

Early life is an essential period for metabolic development, and it is hypothesized that disruptions to microbiome colonization during this period can lead to changes in body composition or susceptibility to metabolic disorders [7]. Murine models support this hypothesis, showing that early life administration of subtherapeutic antibiotic doses to young mice not only changed the composition of their intestinal microbiome (increased ratio of Firmicutes to Bacteroidetes) but also increased their adiposity compared with controls [5,40]. Moreover, the mice were particularly vulnerable to antibiotic exposure during a critical time window around birth [5]. Pups whose mothers were treated with penicillin before delivery and throughout the weaning process had increased adiposity that lasted into adulthood compared with pups who received penicillin after weaning [5]. The results suggest that even transient changes in the microbiota caused by limited antibiotic exposure during a specific time in development may have a long-term effect on body composition. Further, the investigators were able to induce adiposity into GF mice by exposing them to the antibiotic-moderated fecal microbiota, suggesting a causal role for the microbes [5]. Accordingly, early life antibiotic exposure in humans has been associated with an increased risk of being overweight and obese later in childhood and into adulthood [41,42,43]. In addition, long-term antibiotic exposure in childhood has been associated with an increased risk of developing type 2 diabetes [25].

A potential mechanism by which the microbiota may impact host metabolism is via the SCFAs they produce, which are detected at elevated levels within the intestine of obese individuals [44]. Indeed, in the murine models with early life antibiotic-induced adiposity, there were changes in key microbial genes involved in the metabolism of carbohydrates to SCFAs and increased colonic SCFA levels [40]. SCFAs directly provide energy to colonocytes, and increased absorption stimulates adipogenesis [45]. Due to this overproduction in early life, the ecosystem might not be able to recover, and it may continue to have increased harvesting of energy. Another commonly proposed mechanism is through reduced deconjugation of bile acids by intestinal microbiota with an altered fecal bile acid pool being associated with decreased insulin sensitivity, indicating metabolic dysregulation [44].

## 3. C-Section Delivery Disruption of Infant Microbiome Colonization

The C-section delivery rate has been increasing worldwide and, in the United States, has steadily increased over the years from 20.7% in 1996 to 31.8% in 2020 [46,47,48]. Although C-section delivery can be life saving for both mother and infant, it represents a major disruptor of mother to infant microbiome transfer and early life infant microbial colonization.

C-section-delivered infants have decreased maternal to infant microbiome transfer of the maternal vaginal and perineal/fecal microbiome as they bypass the vaginal canal during a C-section delivery [2,3,49]. Dominguez-bello et al. showed that shortly after birth, vaginally delivered infants typically acquire bacterial communities resembling their own mother’s vaginal and perineal microbiota, including *Lactobacillus* spp., while C-section-delivered infants often harbor bacterial communities similar to those found on the skin surface, dominated by *Staphylococcus*, *Corynebacterium*, and *Propionibacterium* spp. [2]. Early in life, the infant gut microbiome of C-section-delivered infants also has decreased transmission from the maternal fecal microbiome, with decreased sharing of *Bifidobacterium* spp. and other taxa [3,7]. In the first few days of life, infants delivered by C-section have been shown to have a decreased abundance of Escherichia and overall increased alpha diversity of the gut microbiome compared to those born vaginally [49,50]. Rapidly, over the first few weeks of life, infants born by vaginal delivery start to develop a higher alpha diversity, with higher levels of *Bifidobacterium* spp. and *Bacteroides* spp. compared to their C-section-delivered counterparts, which help promote development and maturation of the immune system [6,49,51,52]. Infants delivered by C-section have also been demonstrated to harbor higher levels of potential opportunistic pathogens in their guts, including *Klebsiella* spp. [49].

Differences in the gut microbiome by delivery mode, particularly in regards to increased levels of *Bifidobacterium* spp. and *Bacteroides* spp. in vaginally delivered infants, are most prominent within the first 6 months of life and then start to converge, especially after the cessation of breastfeeding and the introduction of solid food [7,53]. Additionally, we have recently shown that gut bacteriophages, which can alter gut microbiome composition and impact host immune responses, also differ by delivery mode in terms of diversity, specific phages, and function, principally in the first few months of life [54]. Although these differences in the gut microbiome by delivery mode may be short-lived, murine models have shown that even transient changes during the critical early life period can have long-lasting inflammatory health consequences [5]. Moreover, some studies have shown that certain individual taxa still differ by delivery mode further into childhood [55,56].

It is important to recognize that additional confounding factors may influence the observed impact of delivery mode on the gut microbiome, with peripartum antibiotic administration being particularly significant, as it is routinely administered during C-section deliveries. Studies have shown that peripartum antibiotics given during a vaginal delivery alters the very early life microbiome, resulting in a lower Bacteroides profile compared to vaginally delivered infants whose mothers do not receive peripartum antibiotics [49,50]. Therefore, the effects of peripartum antibiotics in a C-section delivery are difficult to extricate and may be additive. In addition, there are differences in the gut microbiome of C-section-delivered infants depending on whether the C-section is performed pre- or post-labor, with higher sharing of the maternal vaginal microbiome in infants delivered by C-section post-labor [3]. Moreover, breastfeeding is known to impact the infant microbiome and have a positive influence on offspring health [6,7,57,58]. Breast-fed infants have slower maturation of the gut microbiome than formula-fed infants, with increased levels of taxa that are sometimes used as probiotics such as L. johnsonii/L.gasseri, L. paracasei/L. casei, and B. longum [6,7]. Importantly, breastfeeding can mitigate some of the C-section-associated disruptions to gut microbiome development [58,59]. When receiving breast milk, the gut microbiome differed less between delivery modes compared to when receiving formula [58,59]. Recently, “auxiliary” seeding pathways have been described, where more beneficial breast milk microbiota can colonize the guts of C-section-delivered infants compared to vaginally delivered infants [3,60]. This may be a partial compensatory mechanism to restore the disrupted microbiota following a C-section delivery.

Given that C-section delivery causes aberrations in the development of the gut microbiome, and the developing gut microbiome directly educates the immune system, it is thought that a C-section alters immune development via microbiome-mediated mechanisms. A few studies have tried to address this. One study found that C-section-delivered infants have a lower abundance of Escherichia, Bifidobacterium, and Bacteroides and also have lower serum levels of IgG and IL-12p70 and decreased proportions of IFN-γ/IL-4 compared to vaginally delivered infants [61]. This could potentially lead to an imbalance of Th1/Th2 cells in C-section-delivered infants, although in this study, the authors were not able to directly show that Th1/Th2-cell imbalances in infants born by C-section correlated with reduced abundance of specific microbiota [61]. Another small observational study indicated early differences in the immune system based on delivery mode that were microbiome-mediated, where functional profiles from sequencing revealed enrichment in the lipopolysaccharide (LPS) biosynthesis pathways in the stool of vaginally delivered babies at 5 days of life [62]. LPS forms part of the outer membrane of Gram-negative bacteria and is a highly potent innate immune activator recognized by the Toll-like receptor 4. This was likely due to specific strains of bacteria transferred from the mother to the infant during a vaginal delivery including Bacteroides vulgatus and other Bacteroides spp. Moreover, a targeted cytokine analysis of early life plasma showed significantly increased levels of IL-18 and TNF-α in those born by vaginal delivery compared with those born by C-section. Translating this to immune outcomes, it was recently shown that infants delivered vaginally had increased antibody responses against pneumococcal and meningococcal vaccinations, correlating with an increased abundance of Bifidobacterium and Escherichia coli in the first week of life, compared to C-section-delivered infants [63].

While the focus of this review is on the gut microbiome, it should be noted that delivery mode influences the microbiome colonization at different body sites. Notable variations in microbiome composition related to delivery mode have been seen in infant skin, nasopharyngeal, and oral microbiomes [2,3,64]. Additionally, it has been shown that vaginal delivery provides skin colonization resistance from environmental microbes early in life [65].

## 4. C-Section-Associated Adverse Health Outcomes

There is strong epidemiological evidence that C-section-delivered infants are at increased risk for a wide range of chronic inflammatory, metabolic, and neurodevelopmental conditions compared to infants born by vaginal delivery [9]. The associations of these adverse health outcomes with C-section delivery are at least partially thought to be microbiome-mediated, given evidence from known early life microbiomes and subsequent immune disruption with a C-section delivery from a large longitudinal cohort study with repeated measures and by transfer of these disease phenotypes, including obesity, using microbiota transplantation into GF murine models [5,66].

### 4.1. C-Section Association with Obesity and Metabolic Regulation

Numerous studies, systematic reviews, and meta-analyses have reported an association between C-section delivery, higher BMI, and risk of developing obesity in both childhood and adulthood, after adjusting for obesogenic risk factors [66,67,68,69,70]. Even within the same family, it has been shown that individuals born by C-section had 64% higher risk of obesity than their siblings born vaginally [70]. These associations are consistent across sexes and stronger among younger maternal age and in a C-section without labor compared to a C-section with labor [67,71]. Vu et al. observed that the effect of a C-section on increased infant BMI was predominately associated with a higher Enterobacteriaceae/Bacteroidaceae ratio at 3 months [66]. To a much lesser extent, increased infant BMI was also mediated by Clostridium difficile–Ruminococcaceae pathways at 3 months and lower Bifidobacterium at 12 months [66]. C-section delivery has also been associated with metabolic dysfunction and an increased risk of metabolic syndrome and type 2 diabetes in adulthood, with adolescents delivered by C-section already showing some markers of higher insulin resistance [72,73,74].

### 4.2. C-Section Association with Immune Disorders

Many epidemiological studies have found an association between C-section delivery and an increased risk of atopy, including allergic rhinitis, asthma, allergies, and atopic dermatitis [9,75,76,77,78,79,80]. It should be noted that while the findings from individual studies have not always been consistent, several of the associations have been supported by either meta-analyses or very large observational studies. Vu et al. showed that higher Enterobacteriaceae/Bacteroidaceae levels at 12 months was the main contributor to atopy risk at 1 and 3 years, with the abundance of Veillonellaceae and C. difficile at 12 months and Enterobacteriaceae/Bacteroidaceae ratio and gut microbiome-derived formate at 3 months contributing as well but to a lesser extent [66].

Stokholm et al. discovered an association between C-section-altered gut microbial composition and an increased asthma risk up to the first 6 years of life [77]. In an additional study examining asthma, C-section-delivered infants displayed a lower abundance of Actinobacteria and Bacteroides at 1 week, with Bacteroides continuing to be lower at 1 month [77]. In comparison, Firmicutes and Proteobacteria abundances were both higher at 1 week and 1 month among C-section-delivered infants. The majority of these delivery mode-associated changes were resolved by 1 year among most infants; however, infants whose microbiome changes were sustained at 1 year of age were found to have increased risk for asthma [77]. Researchers in this study constructed a C-section microbial score that evaluated the level of gut microbial alteration among different taxa. A higher C-section-influenced microbial score at age 1 consisted of lower abundance of *Bacteroides*, *Faecalibacterium*, *Roseburia*, *Ruminococcus*, *Akkermansia*, *Alistipes*, and *Dialister* and a higher abundance of *Clostridia*, *Enterobacteriaceae* genus, *Veillonella*, *Bifidobacterium*, and *Lachnospiraceae* genus. Among 549 vaginally delivered infants, 6% developed asthma by age 6. However, in 151 C-section born infants, among the infants with higher C-section-influenced microbial scores, 20% developed asthma by age 6, compared to only 7% in the group with lower C-section-influenced microbial scores [77]. Children with a higher C-section-influenced microbial score at 1 year of age were also observed to have lower airway immune mediator responses [77]. This suggests that long-term disrupted maturation of the infant gut microbiome increases asthmatic risk, but a healthy maturation of an altered gut composition can reduce risks. In addition, C-section delivery is associated with an increased number of respiratory tract infections in early life, which increased the risk of asthma development in itself [81,82,83].

In a cohort of 112 Chinese children, Cheung et al. observed alterations in gut microbial composition and diversity during the first 3 years of life, associated with varying early life factors, including birth mode and the presence of eczema [84]. They showed an increased abundance of *Finegoldia magna*, which is associated with excessive T-cell cytokine production and weakening host immune defense, in infants with eczema at one month of age [84]. In 1-year-old infants with eczema, they observed a reduction in Bacteroides during 1 to 6 months of life and a higher abundance of Clostridium sensu stricto 1 during 1 to 3 months of life, which is a shift seen in C-section-delivered infants [84].

Epidemiological studies have also shown associations with C-section and other inflammatory diseases, although to a lesser extent and often reporting conflicting data between studies or in meta-analyses. These include autoimmune diseases such Celiac disease, type 1 diabetes, inflammatory bowel disease, systemic connective tissue disorders, and juvenile arthritis [85,86,87,88,89]. In addition, C-section delivery has been associated with certain childhood cancers, including lymphomas, leukemia, and sarcoma, and also early onset colorectal cancer [89,90,91].

### 4.3. C-Section Association with Neurodevelopmental Disorders

C-section delivery has been associated with a variety of neurodevelopmental disorders in some studies. Lower motor and language development scores have been seen during the first three years of life in infants delivered by C-section [92]. A meta-analysis showed that C-section delivery was associated with increased odds of autism spectrum disorders (Odds Ratio 1.33) and attention-deficit/hyperactivity disorder (Odds Ratio 1.17) compared with vaginal delivery, with comparable estimates irrespective of whether the C-section was elective or an emergency [93]. Higher odds were also seen for other neurodevelopmental disorders, including learning disabilities, tic disorders, obsessive–compulsive disorder, and eating disorders; however, they were not statistically significant [93]. Decreased white matter in the brain in early infancy has also been seen in infants delivered by C-section compared to vaginally delivered infants [94]. Although early life gut microbes and metabolites have been associated with the development of future neurodevelopmental disorders, to our knowledge, these have not yet been associated with C-section delivery directly [95,96].

Despite epidemiological evidence showing an association with C-section delivery and an increased risk of some metabolic, immune, and neurological disorders, it is also possible that certain factors inherently associated with C-section delivery may predispose to these conditions. For example, delayed initiation of breastfeeding commonly occurs with C-sections and can possibly impact the risk of C-section-associated diseases [97]. Additionally, C-section delivery is more common in pregnant women with obesity, which in itself is a risk factor for offspring obesity [98,99]. When evaluating these epidemological studies, it should be assessed whether possible confounding factors were accounted for.

### 4.4. Rationale for Restoring the Microbiome of C-Section Infants by Vaginal Seeding

Given the increase in C-section rates worldwide and the increasing recognition of C-section-associated health conditions that are at least in part microbiome-mediated, there has been growing interest in trying to “restore” the microbiome of C-section-delivered infants to improve their health outcomes. One method of restoration has been dubbed “vaginal seeding”, a process by which C-section-delivered infants receive exposure to their own mothers’ vaginal microbes, which they would have otherwise been exposed to during a vaginal delivery [10].

Traditionally, vaginal seeding refers to the process first described by Dominguez-Bello et al. in which a sterile gauze is inserted into the vagina for approximately an hour prior to the C-section procedure. The gauze is then removed from the vagina before the delivery. In some studies, this is prior to the mother receiving peripartum antibiotics for their C-section prophylaxis, and in others, this is after receiving peripartum antibiotics [10,100,101]. As soon as the infant is stable after delivery, the gauze is wiped over the infant’s mouth, face, and body, before the infant is placed skin-to-skin with their mother (Figure 1) [10].

Studies have shown that this intervention partially restores a C-section-delivered infants’ microbiome, shifting it closer to a vaginally delivered infant than a C-section delivered infant without vaginal seeding [10,101]. Vaginal seeding and its associated microbiome changes are hypothesized to help normalize the infant’s immune system development and metabolic regulation, subsequently reducing their risk for C-section-associated inflammatory conditions and metabolic dysregulation (Figure 2) [102].

## 5. Objectives

A scoping review was conducted in order to systemically map the research conducted in vaginal seeding of C-section-delivered infants, as well as to identify any existing gaps in the knowledge. The following research question was formulated: What is known from the literature about the efficacy of vaginal seeding on the infant microbiome and infant health outcomes, in addition to challenges, safety concerns, and regulatory issues?

## 6. Methods

Our protocol followed the PRISMA-SCR guideline for scoping reviews to the extent possible for this topic [103]. To be included in the review, papers needed to focus on vaginal seeding or other forms of maternal–infant microbial seeding of C-section-delivered infants. Peer-reviewed journals were included if they were published between 2010 and 2025 and involved human participants or the use of human samples in models. Papers discussing regulatory and safety issues were also included. To identify potentially relevant articles, the following bibliographic databases were searched from 2010 to April 2025: MEDLINE and EMBASE. Sources of evidence were screened by at least three authors before a decision was made on whether they met criteria to be included. Data from eligible studies were charted using a standardized data extraction tool used for this study, including microbiome changes with vaginal seeding and other maternal–infant microbial interventions, and health outcome effects of vaginal seeding on C-section-delivered infants.

## 7. Evidence of the Effect of Vaginal Seeding and Other Maternal–Infant Microbial Seeding Interventions on the Infant Microbiome

A handful of studies have examined the effects of vaginal seeding and other maternal–infant microbial seeding interventions on the microbiome of a C-section-delivered infant (Table 1). While most studies have shown a shift in the infant microbiome with seeding, results have varied depending on several factors, including seeding method and study design.

In the initial pilot observational study by Dominguez-Bello et al., vaginal seeding was found to shift the microbiome composition in the first month of life of a seeded infant closer to that of a vaginally delivered infant compared to a C-section-delivered infant who did not receive seeding [10]. Specifically, seeded infants had enriched *Lactobacillus* and *Bacteroides* in their anal swabs and enriched *Lactobacillus* and Bacteroidales in their skin swabs. Extending this study to a larger cohort, the differences in microbial composition observed in vaginally seeded infants remained for up to a year in the gut and skin microbiome, with specific taxa differences seen in the gut, skin, and oral microbiome for up to 6 months [101].

Randomized controlled trials (RCTs) have also shown that vaginal seeding can shift an infant’s microbiome [100,104]. A double-blind RCT in vaginal seeding demonstrated increased maternal bacterial engraftment in the infant gut and skin over the first month of life in seeded infants [100]. Further, this study observed decreased alpha diversity in the transitional stool of seeded infants, which is consistent with the early life lower stool diversity associated with vaginally delivered infants. This study also showed that vaginal seeding reduced potential pathobionts in the infant gut and skin, including Enterobacteriaceae in the stool. Differences in microbiome composition with vaginal seeding and a reduction in pathobionts were also noted in another RCT [104]. An additional RCT demonstrated a trend of increased *Lactobacillus* and *Bacteroides* in the infant gut at day 1 and 6 months in infants who had received vaginal seeding; however, this did not reach statistical significance [105]. Of note, vaginal seeding has also been shown to influence the gut metabolome, with one study showing enrichment of several carbohydrates (D-xylulose, D-ribulose, beta-D-fucose, N-acetyl-D-glucosamine, tartaric acid, and rhamnose), amino acids (2-phenylglycine and 3-chlorotyrosine), chenodeoxycholic acid, L-lactic acid, carnosine, hydroxyphenyllactic acid, homovanillic acid, acetic acid, and indolelactic acid [104]. In the future, an assessment of the specific microbial taxa or groups of taxa driving beneficial immune or metabolic changes in vaginally seeded infants will be important to examine. Currently, this is challenging given the lack of consistency in study design and lack of correlation with health outcomes in studies to date.

Approaches other than vaginal seeding for maternal–infant microbial seeding of C-section-delivered infants have been proposed. As gut microbial colonization is highly important for immune and metabolic regulation, Wilson et al. examined whether giving maternal vaginal fluids by oral administration to a C-section–delivered infant may be effective [106]. In this pilot trial, no changes in the gut microbiome of C-section-delivered infants were observed at 1 or 3 months in those receiving vaginal fluid orally compared to those receiving a placebo. This may have possibly been due to dilution of the vaginal bacterial load during preparation or the study being underpowered.

There have been criticisms questioning the utilization of vaginally sourced microbes in colonizing the diverse microbiome body sites of an infant [107]. The vaginal microenvironment is acidic and typically dominated by *Lactobacillus* species. In contrast, the microenvironment of the infant mouth, skin, and gut offers varied and differing conditions compared to the vagina. In addition, studies comparing gut microbiome differences between C-section and vaginally delivered infants repeatedly show increased *Bacteroides* in the guts of vaginally delivered infants; however, vaginal seeding has not consistently shown an increase in *Bacteroides* colonization between all studies despite causing other microbiome changes. For this reason, a proof-of-concept study investigated the use of maternal stool to specifically restore the infant gut microbiome environment [108]. Korpela et al. collected maternal stool samples three weeks prior to C-section delivery and administered diluted maternal feces with breast milk to seven infants as their first feed [108]. This resulted in an increase in *Bacteroides* up to 21 days and Bifidobacteriales up to 3 months in the stool and a reduction in pathobionts. This suggests that maternal gut bacteria may also play an important role in infant gut colonization. Consistently, it has been shown that the vaginal microbiome, which is usually in a steady state enriched with *Lactobacillus* sp., towards the end of pregnancy diversifies to include taxa usually associated with other body sites, notably the gut, with maternal fecal and vaginal microbiomes becoming more similar later in pregnancy [100,101,109]. We theorize this diversification physiologically allows the vaginal flora to “seed” different sites of the infant’s body with the necessary microbiome during a vaginal delivery.

Additionally with vaginal seeding, the number of vaginal microbes that are carried on the gauze and reach the infant is unclear. Some studies show that the incubated gauze contains microbes similar to those of vaginal swabs, with about 75% of bacterial amplicon sequence variants from the swab also present in the gauze [10,101]. Interestingly, one study found that only a little over a quarter of the vaginal microbes in the gauze were viable when analyzed via flow cytometry [106]. These findings indicate that collection via gauze may be an adequate practice to collect a representative vaginal microbiota sample but may fail to maintain the viability of those vaginal microbes. This requires further investigation to understand which microbes may be selected by the gauze collection process and the role of cell viability in successful vaginal seeding intervention. It is unclear if viable microbes are essential in influencing colonization or if the non-viable components (e.g., dead bacteria and metabolites) of the vaginal fluid are sufficient to shape the early life microbiome. In addition, the optimal length of time of incubation of the gauze in the vagina, how long the gauze can be removed from the vagina, and how it should best be stored before swabbing the infant to ensure sufficient microbial transfer still need to be explored.

**Table 1 microorganisms-13-01236-t001:** Current research on microbiota changes in C-section infants as the result of vaginal seeding and other early life maternal-derived microbial interventions.

Author Location Date Published	Title	Study Patients	Sample Collection	Sequencing Type	Diversity	Enriched Bacteria in Seeded Infants	Reduced Bacteria in Seeded Infants	Key Finding and Contributions
Dominguez-Bello MG et al. USA, Puerto Rico March 2016	Partial restoration of the microbiota of Cesarean-born infants via vaginal microbial transfer [10]	**4 VS** 7 CS 7 VD	Birth, 3 d, 7 d, 14 d, 21 d, 30 d (oral, skin, and anal swabs)	16S RNA	No difference between groups	**Anal:** *Lactobacillus* (7 d), *Bacteroides* (14 d) **Skin:** *Lactobacillus* (overall), Bacteroidales family S24–7 (overall)	Not Reported	-VS infant microbiome more similar to VD than CS
Mueller NT et al. USA June 2023	Maternal Bacterial Engraftment in Multiple Body Sites of Cesarean Section Born Neonates after Vaginal Seeding—a Randomized Controlled Trial [100]	**10 VS** 10 placebo-CS	1 d (stool and skin), 30 d (stool)	16S RNA	**α:** VS reduced diversity in stool (1 d and 30 d) and skin (1 d) **β:** significant variance in stool and skin driven by treatment (1 d)	**Gut:** *Aeromonas* (1 d), *Stenotrophomonas* (30 d), *Ruminobacter* (30 d) **Skin:** *Lactobacillus*, *Parvimonas*, Ruminococcaceae (1 d)	**Gut:** *Enterobacter* (1 d), *Enterobacteriaceae* (1 d), *Clostridium sensu stricto* (30 d) **Skin:** *Streptobacillus*, *Lactococcus*, *Acidibacter* (1 d)	-First double-blinded, randomized, placebo-controlled trial of VS
Song SJ et al. USA, Chile, Bolivia, Spain August 2021	Naturalization of the microbiota developmental trajectory of Cesarean-born babys after vaginal seeding [101]	**30 VS** 49 CS 98 VD	Birth, 1 d, 2 d, 7 d, 14 d, 21 d, and monthly up to 1 yr (oral, skin, and stool swabs)	16S RNA	**α:** no difference **β:** significant difference between groups	**Gut:** *Bacteroides*, *Streptococcus*, *Clostridium* (up to 6 m) **Skin:** *Streptococcus*, *Neisseria*, *Thermus*, *Neisseriaceae* (up to 6 m) **Oral:** *Gemellaceae*, *Haemophilus*, and *Streptococcus* (up to 6 m)	VS not effective in reducing CS-associated taxa	-VS developed more similarly to VD for gut and skin microbiome, not oral -VS microbiome was more stable than CS during first year of life
Zhou L et al. China July 2023	Effects of vaginal microbiota transfer on the neurodevelopment and microbiome of Cesarean born infants: A blinded randomized controlled trial [104]	**32 VS** 36 placebo-CS 33 VD	3 d, 7 d, 30 d, 6 wk (stool)	16S RNA	**β:** VS was significantly different from placebo-CS group but not VD group (42 d)	**Gut:** *Lactobacillus* (3 d to 6 wk)	**Gut:** *Klebsiella* (decreasing trajectory at 30 d and 6 wk), *Bifidobacterium* (increasing trajectory, after 7 d), *Escherichia* (increasing trajectory, after 30 d)	-VS accelerated gut microbiome maturation compared to placebo
Wilson BC et al. New Zealand June 2021	Oral administration of maternal vaginal microbes at birth to restore gut microbiome development in infants born by caesarean section: A pilot randomised placebo-controlled trial [106]	**12 oral-VS** 13 placebo-CS 22 VD	Birth, 1 m, 3 m (stool)	Shotgun Metagenomic Sequencing^1^	No significant difference between oral-VS and placebo-CS groups	No significant difference between oral-VS and placebo-CS groups	No significant difference between oral-VS and placebo-CS groups	-Maternal vaginal strain transmission was present in 4/12 oral-VS at 1 m and only 1/12 at 3 m -Oral-VS ineffective at influencing early life microbiome
Korpela K et al. Finland October 2020	Maternal Fecal Microbiota Transplantation in Cesarean-Born Infants Rapidly Restores Normal Gut Microbial Development: A Proof-of-Concept Study [108]	**7 FMT** 18 CS 29 VD	Birth, 2 d, 7 d, 14 d, 21 d, 28 d, 3 m (stool)	16S RNA	No significant difference between groups	**Gut:** *Bacteroides* (up to 21 d), Bifidobacteriales (21 d to 3 m)	**Gut:** potentially pathogens including *Enterococcus faecium*, *Enterococcus faecalis*, *Klebsiella pneumoniae*, *Salmonella enterica* (7 d, 3 m)	-FMT gut microbiome was more similar to VD -FMT overcame lack of *Bacteroides* and delayed *Bifidobacteria* characteristic of CS
Liu Y et al. China January 2023	Effects of vaginal seeding on gut microbiota, body mass index, and allergy risks in infants born through Cesarean delivery: a randomized clinical trial [105]	**57 VS** 60 placebo-CS	Birth, 6 m, 12 m, 18 m, 24 m (stool)	16S RNA	No significant difference between groups	**Gut:** *Lactobacillus*, *Bacteroides* (not significant trend, birth and 6 m)	No significant difference between groups	-Gut microbiome composition did not vary significantly between VS and CS infants

VD (vaginally delivered infant), CS (C-section infant), VS (vaginal seeded infant), FMT (fecal microbiota transplant infant), d (days), wk (weeks), m (months), yr (years), 16S (16S ribosomal RNA gene sequencing). No significant adverse events were reported in any study. Treatment groups are indicated with bold font in the “Study Patients” column.

## 8. Evidence of the Effect of Vaginal Seeding and Other Maternal–Infant Microbial Seeding Interventions on the Infant Microbiome

There is clear evidence from observational studies and now RCTs that vaginal seeding can change the composition of a C-section-delivered infant’s microbiome; however, the question remains of whether vaginal seeding can mitigate the microbiome-mediated C-section-associated adverse health outcomes [110,111]. To date, only a few studies have examined this (Table 2).

The first evidence of the benefit of vaginal seeding on health outcomes was recently published, examining neurodevelopmental outcomes [104]. In this triple-blind RCT, C-section-delivered infants that received vaginal seeding had significantly higher overall neurodevelopmental scores at 3 and 6 months, as measured by the Ages and Stages Questionnaire (ASQ-3), than C-section-delivered infants receiving a placebo. Specifically, they had higher communication scores at 3 months and gross motor scores at 3 and 6 months [104]. It should be noted that even though the infants who received the placebo had lower neurodevelopmental scores, these scores were still not generally in the range that would flag assessment for neurodevelopmental delay.

To date, there have been no studies showing that vaginal seeding improves inflammatory and metabolic outcomes for C-section-delivered infants. A small RCT conducted in China compared the health outcomes of C-section-delivered infants who received vaginal seeding versus routine standard of care, following participants up to the first two years of life [105]. No differences were seen in allergy scores or body mass index (BMI) between the two groups. The study may have been underpowered to detect BMI differences (n = 60 in each arm) and may have needed to follow the infants longer into childhood to see a differences in outcomes. However, a recent human-to-murine model suggests a possible improvement in metabolic health with vaginal seeding [112]. In this study, GF mice were inoculated with transitional stool from C-section-delivered infants who received vaginal seeding versus a placebo. The mice that received stool from vaginally seeded infants showed decreased intraabdominal adipose tissue compared with mice that received stool from infants receiving a placebo. No differences were seen in other measures of adiposity such as weight, weight gain trajectories, epididymal tissue fat, and subcutaneous tissue volume. Of note, intraabdominal adiposity, compared to peripheral adiposity, is more strongly associated with adverse metabolic outcomes including insulin resistance, metabolic syndrome, and cardiovascular disease in humans [113]. This suggests that in human studies of vaginal seeding, other markers of adiposity rather than just BMI may need to be assessed.

It is now critical to assess from well-powered, robustly designed RCTs whether vaginal seeding can improve the inflammatory and metabolic outcomes of C-section-delivered infants. To our knowledge, there are currently three ongoing clinical trials studying vaginal seeding of C-section-delivered infants examining obesity and atopy health outcomes: (1) NCT03298334, in VA, USA, with a primary outcome of adiposity; (2) NCT03567707, in NY, USA, with a primary outcome of sensitization to food allergens; and (3) NCT0392843, in Stockholm, Sweden, with a primary outcome of allergic diseases and also exposing infants to their mother’s fecal microbiota in addition to vaginal microbiota.

## 9. Safety and Challenges

As more studies are conducted to assess the effects of vaginal seeding on the infant microbiome and associated health outcomes, there also needs to be an assessment on the safety and challenges associated with this technique. In 2017, The American College of Obstetricians and Gynecologists (ACOG) stated “At this time, vaginal seeding should not be performed outside the context of an institutional review board (IRB) approved research protocol until adequate data regarding the safety and benefit of the process become available” [114].

The main safety concern with vaginal seeding is the risk of transmission of infection as the C-section-delivered infant is exposed to the maternal vaginal fluid, which may harbor infectious agents that they would have otherwise bypassed. Extensive screening for potential pathogens is important, along with data safety and monitoring to ensure patient safety, as has occurred in many of the published and ongoing clinical trials in vaginal seeding [115]. In our current National Institute of Allergy and Infectious Diseases-sponsored RCT in vaginal seeding (NCT03298334), mothers are screened for sexually transmitted infections (STIs) including Gonorrhea, Chlamydia, Hepatitis B, Hepatitis C, Syphilis, and Human Immunodeficiency Virus at 35 weeks gestation or later in addition to the requiring negative standard of care testing for STIs earlier in pregnancy. Testing for Group B Streptococcus is also performed at 35–37 weeks gestation. Additionally, vaginal pH is tested prior to insertion of the vaginal gauze and must be ≤4.5 to ensure a Lactobacillus-dominated vaginal microbiota and lower chance of bacterial vaginosis. Furthermore, women are excluded if their C-section is scheduled for an active infection that would have interfered with vaginal delivery or if they have symptoms of a vaginal infection on delivery admission such as genital herpetic lesions. False negatives, acquisition of pathogens after testing is complete, and asymptomatic infections are also important to take into consideration. For example, approximately 60–80% of women who deliver a Herpes Simplex Virus (HSV)-infected infant have no evidence of genital HSV infection at the time of birth, along with no genital herpes history among themselves or their partner [116]. In addition, the vaginal microbiome can harbor other potential pathogens which are not tested for in clinical trials, including *Escherichia coli* containing the K1 antigen, which may cause invasive disease in vulnerable newborns [117]. Furthermore, fecal transplant studies have shown that even healthy individuals can harbor low levels of pathogenic organisms which can become disease causing when transferred to a vulnerable individual, thus resulting in increased infectious screening required for fecal transplant [118].

To date, no infection transmission has been reported with vaginal seeding in approved clinical trials, and vaginal seeding appears safe [104,105]. A case of HSV vesicular lesions over the eyelids was reported in an infant after a vaginal seeding procedure that was performed outside of an IRB-approved research protocol [116]. There was no way to definitively assess if vaginal seeding was the cause as the gauze was not kept for testing of infectious agents; in most regulated clinical trials, the gauze is required to be stored. Additionally, the mother had previously reported a history of “occasional oral cold sores” but had no reported history of genital herpes or oral HSV lesions since the birth of the infant. Nevertheless, this emphasizes that for now, vaginal seeding should only be performed under approved clinical trials where infections are carefully screened for and safety data followed. Careful attention should be given to infectious diseases that are acquired that are not in the current screening protocols, to assess if they were transmitted via vaginal seeding and should be added to screening protocols. This also emphasizes the need to store study materials such as the vaginal gauze used for seeding to be able to access for possible causation.

Another challenge of vaginal seeding is the variability of the vaginal microbiome between mothers. Human-to-mouse studies of vaginal seeding with human vaginal microbial communities suggest that the composition of the transferred seeding material is important in subsequent infant immune and metabolic development, which can be modified by different factors, including prenatal maternal health [119]. Given this, vaginal seeding may not be equally effective from all mothers. Ongoing and future trials should be large enough to account for this variability in microbiome and potential effects and to be able to assess which components of the microbiome may be the most effective in improving health outcomes. Additionally, postnatal factors such as delayed initiation of breastfeeding, which is common with C-sections, can possibly impact the risk of C-section-associated diseases [97,120]. Moreover, breastfeeding itself can positively impact the infant gut microbiome and diminish microbiome differences seen by delivery mode [6,7,58,59]. Robust, well-powered trials with randomization so these factors can be balanced in each trial arm are needed to account for this. Lastly, current ongoing clinical trials often have very stringent inclusion and exclusion criteria, to provide the “healthiest” microbiome and minimize the risk of infections, with many pregnant women scheduled for C-sections screen failing. Therefore, if vaginal seeding is found to be effective in improving infant health outcomes, whether this can be generalizable to all C-section-delivered infants would need to be assessed.

## 10. Regulatory and Ethical Issues Around Vaginal Seeding

Clinical trials, particularly ones involving a new intervention or drug, have specific regulations and safety measures, as determined by the US Food and Drug Administration (FDA), and often require an Investigation New Drug Application (IND). The FDA has not issued public guidance on vaginal seeding, and requests by investigators are handled on a case-by-case basis [121]. Both ongoing US clinical trials studying vaginal seeding are being conducted under an IND based on the FDA’s determination that the application of the maternal vaginal flora to infants to determine impact on health outcomes in children “utilizes administration of live organisms to affect structure/function of the body” [121]. Whether “maternal vaginal flora” meets the definition of a “drug/biologic” under the federal Food, Drug and Cosmetic Act remains largely debated. Vaginal fluid is heterogeneous and varies between pregnant women, so is fundamentally difficult to characterize, unlike many of the other drugs under FDA oversight, with the exception of fecal microbiota transplantation (FMT). FMT, the transfer of feces from a healthy donor to a recipient, is under the purview of the FDA; however, it can be argued that the same regulations for FMT are not necessary for vaginal seeding. With FMT, the recipient would have never otherwise been exposed the donor’s fecal material, whereas in vaginal seeding, the infant would have been naturally exposed to the maternal vaginal fluid during a vaginal delivery, but a C-section medical intervention interrupts this process [121]. Regardless, the additional oversight provided by the FDA and IND ensure rigorous assessment of efficacy and safety, which is much needed, as vaginal seeding is still being investigated. To our knowledge, official guidelines regarding the use of vaginal seeding are not available in other countries to date. However, trials outside the USA, including NCT0392843 in Stockholm, Sweden, and a trial in New Zealand, follow similar screening criteria for infectious agents as trials conducted within the USA [106].

## 11. Conclusions and Future Directions

Studies to date, including RCTs, indicate that vaginal seeding can partially restore the microbial colonization of C-section-delivered infants closer to that of vaginally delivered infants. However, knowledge of the impact of this microbial intervention on C-section-associated diseases is very limited, with a paucity of well-designed studies, with efficacy only seen for early life neurodevelopmental outcomes to date [104]. Large, rigorous RCTs are needed and are currently underway to assess the impact of vaginal seeding on inflammatory and metabolic outcomes, as well as the safety of the process. When these are complete, it may enable a systematic review with or without a meta-analysis to be conducted. If found to be efficacious, trials are needed to evaluate the inclusion and exclusion criteria to assess whether they are generalizable to all C-section-delivered infants.

As well as assessing the overall impact of vaginal seeding on health outcomes, given the heterogeneity of the vaginal microbiome, studies should also examine whether there is an “optimal” composition that restores colonization and improves health outcomes. Knowledge of the most efficacious combination of microbes and their metabolites may allow for the development of a live biotherapeutic product that would be more standardized. This could have utility for C-section-delivered infants who will never be able to receive vaginal seeding from their own mothers due to factors such as maternal infection. However, at this stage, it is unclear how important it is for a C-section-delivered infant to receive the unique microbiome from their own mother versus a generic “optimal” microbiome and warrants further investigation.

We are excited to see the results from ongoing robust RCTs in vaginal seeding and are cautiously optimistic that we will see an improvement in C-section-associated inflammatory and metabolic conditions. If this is the case, eventually, this simple, inexpensive procedure may be incorporated into medical practice to mitigate C-section-associated diseases, potentially improving the health of many and reducing healthcare burden.

## Figures and Tables

**Figure 1 microorganisms-13-01236-f001:**
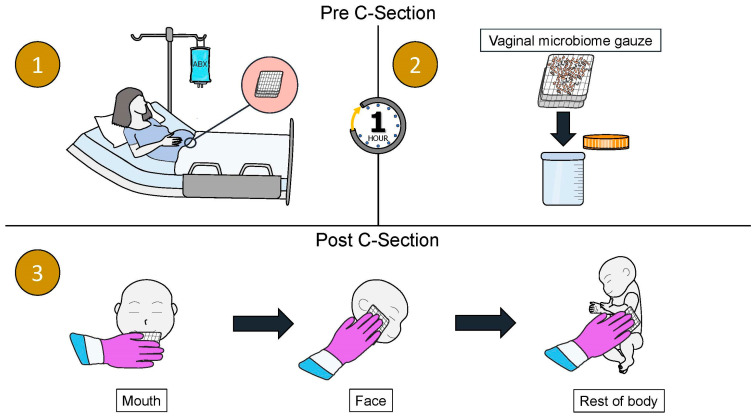
A schematic diagram of the vaginal seeding procedure. (**1**) A sterile gauze is inserted into the mother’s vagina while she is waiting for the C-section procedure for approximately 1 h. (**2**) The gauze is removed from the mother’s vagina prior to the C-section procedure. (**3**) As soon as the infant is stable post-delivery, the gauze is wiped over the infant’s mouth, face, and rest of the body.

**Figure 2 microorganisms-13-01236-f002:**
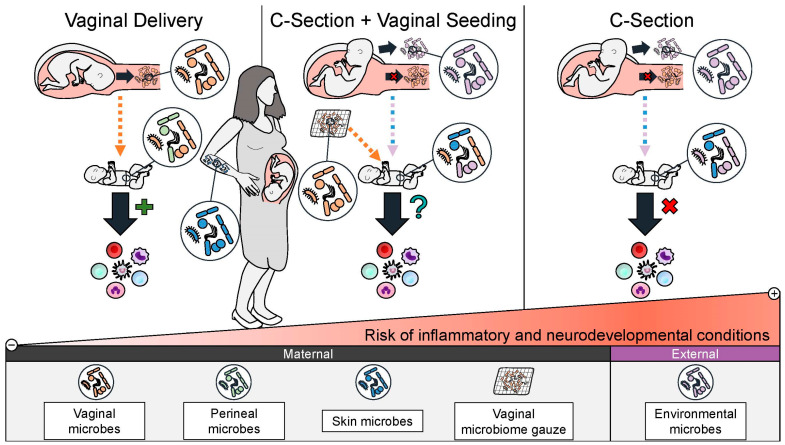
A schematic of the rationale for vaginal seeding to improve health outcomes for C-section-delivered infants. Vaginal seeding shifts the composition of a C-section-delivered infants’ microbiome closer to that of a vaginally delivered infant compared to C-section-delivered infant who did not receive vaginal seeding. It is hypothesized that this may restore aberrant immune system education and metabolic regulation seen in C-section-delivered infants and decrease their risk of future inflammatory, metabolic, and neurodevelopmental conditions.

**Table 2 microorganisms-13-01236-t002:** Studies examining the effects of vaginal seeding on C-section-associated adverse health outcomes.

Authors	Title	Journal Published	Year Published	Location and Patient Population Studied	Key Findings
Zhou L, Qiu W, Wang J, et al.	Effects of vaginal microbiota transfer on the neurodevelopment and microbiome of Cesarean-born infants: A blinded randomized controlled trial [104]	*Cell Host & Microbe*	2023	US, 32 US women of child-bearing age exposed to vaginal seeding, 36 US women of child-bearing age undergoing normal vaginal delivery	Vaginal seeding procedure is associated with improved neurodevelopment in Cesarean-born infants.
Liu Y, Li HT, Zhou SJ, et al.	Effects of Vaginal Seeding on Gut Microbiota, Body Mass Index, and Allergy Risks in Cesarean-Delivered Infants: A Randomized Clinical Trial [105]	American Journal of Obstetrics and Gynecology MFM	2022	China, 60 Chinese infants receiving vaginal seeding, 60 controls	For infants born via C-section, vaginal seeding has no significant impacts on the gut microbiota, growth, or allergy risks during the first 2 years of life.
Namasivayam S, Tilves C, Ding H, et al.	Fecal transplant from vaginally seeded infants decreases intraabdominal adiposity in mice [112]	*Gut Microbes*	2024	US, 8 US healthy infants, 4 receiving vaginal seeding and 4 controls	There was a reduction in IAAT volume in male mice that received stool from vaginally seeded infants compared to control infants.

## Data Availability

No new data were created.

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
