# Peer review of "Can Vaginal Seeding at Birth Improve Health Outcomes of Cesarean Section-Delivered Infants? A Scoping Review"

_microorganisms, 2025, doi:10.3390/microorganisms13061236_

Round 1

Reviewer 1 Report

Comments and Suggestions for Authors

This article presents a scoping review with a broad and engaging topic. However, it lacks the methodological framework typically associated with scoping reviews (refer to Tricco et al.). The results and discussions are highly relevant to clinicians and the community, particularly for mothers and their children. While the study has scientific merit, it is crucial to include a 'Methods' section that clearly outlines the research methodology, including the search steps and selection criteria.

Additional comments are in the attached file.

Author Response

1. Summary

2. Point-by-point response to Comments and Suggestions for Authors

Comments 1: This article presents a scoping review with a broad and engaging topic. However, it lacks the methodological framework typically associated with scoping reviews (refer to Tricco et al.). The results and discussions are highly relevant to clinicians and the community, particularly for mothers and their children. While the study has scientific merit, it is crucial to include a 'Methods' section that clearly outlines the research methodology, including the search steps and selection criteria.

Response 1: Thank you for this important point. We agree with this comment. Therefore, we have added an objectives and methods section for the evidence for vaginal seeing in section 6 and 7 of the manuscript following the PRISMA-ScR checklist [1] (Tricco et al).

Comments from highlighted sections of manuscript:

Comments 2: WHICH: Murine models support this hypothesis, showing that early life administration of subtherapeutic antibiotic doses to young mice not only changed the composition of their intestinal microbiome (increased ratio of Firmicutes to Bacteroidetes) but also increased their adiposity compared with controls [40].

Response 2: Apologies for the lack clarification. We have now referenced both murine models that support this hypothesis to the sentence (references 5 and 40), originally reference 5 was missing.

Comments 3: Line 366: Trials? But the authors cited only 1 reference, please check it.

Response 3: Apologies, references supporting this are now cited directly after this sentence.

Comments 4: Table 1. The table has no vertical lines and only features boxes.

Response 4: On both the word document and PDF we have of the table, vertical lines are shown. This may be an error the reviewer mode for the article. We will ensure that the published version appears in the view we formatted it in, as a full table with vertical lines.

Comments 5: Legend table 1: please verify if there are any extra parentheses

Response 5: For clarity, “d (day(s)), wk (week(s)), m (month(s)), yr (year(s))” has been changed to “d (days), wk (weeks), m (months), yr (years)”

Comments 6: What limitations were present in this study? Could a critical literature analysis be conducted in the future such through a systematic review with or without meta-analysis?

Response 6: This is an important point. We have edited out limitations in the first paragraph of section 12 “Conclusions and Future Directions” to read:

“However, knowledge of the impact of this microbial intervention on C-section associated diseases is very limited with a paucity of well-designed studies, with efficacy only seen for early life neurodevelopmental outcomes to date [104]. Large, rigorous RCTs are needed and are currently underway to assess the impact of vaginal seeding on inflammatory and metabolic outcomes, as well as the safety of the process. When these are complete, it may enable a systematic review with or without meta-analysis to be conducted.”

  1. Tricco, A.C.; Lillie, E.; Zarin, W.e.a. PRISMA Extension for Scoping Reviews (PRISMA-ScR): Checklist and Explanation. Annals of Internal Medicine 2018, 169, 467-473, doi:10.7326/m18-0850 %m 30178033.

Reviewer 2 Report

Comments and Suggestions for Authors

In my opinion, this review requires major revisions. My principal suggestions are as follows:

  1. Table 1: Instead of including the title in the first column, please list the authors.

  2. Microbial taxa: Consider adding more details on whether specific microbial taxa (e.g., Bacteroides, Bifidobacterium) are the main drivers of the beneficial immune or metabolic changes observed in vaginally seeded infants. If a particular genus consistently correlates with improved biomarkers, highlighting this could guide future interventions or screening.

  3. Global regulatory perspective: Include a brief discussion on how international regulatory agencies may view vaginal seeding, as the FDA’s approach is not universally adopted. If European Union guidelines (or others) exist, referencing them would enhance the global relevance of your review.

Author Response

Response to Reviewer 2 Comments

1. Summary

Thank you very much for taking the time to review this manuscript. Please find the detailed responses below and the corresponding revisions/corrections highlighted/in track changes in the re-submitted files that we hope address your comments.

Comments 1: Table 1: Instead of including the title in the first column, please list the authors.

Response 1: Thank you for this suggestion. The authors are now in this first column for Table 1. For consistency we have also made this change for Table 2.

Comments 2: Microbial taxa: Consider adding more details on whether specific microbial taxa (e.g., Bacteroides, Bifidobacterium) are the main drivers of the beneficial immune or metabolic changes observed in vaginally seeded infants. If a particular genus consistently correlates with improved biomarkers, highlighting this could guide future interventions or screening.

Response 2: Thank you, we agree this is important. However, there is a paucity of studies to date that allow us to make firm conclusions on this due to lack of consistency in study design and health outcomes to correlate with microbiome changes. This is now addressed on page 9, line 420 of the manuscript with the addition of the sentences: “In the future, assessment of the specific microbial taxa or groups of taxa driving beneficial immune or metabolic changes in vaginally seeded infants will be important to examine. Currently, this is challenging given the lack of consistency in study design and lack of correlation with health outcomes in studies to date.”

Comments 3: Global regulatory perspective: Include a brief discussion on how international regulatory agencies may view vaginal seeding, as the FDA’s approach is not universally adopted. If European Union guidelines (or others) exist, referencing them would enhance the global relevance of your review.

Response 3: Thank you, this is valuable addition. To address this, we have included the following in section 11 “regulatory and ethical issues around vaginal seeding”:

To our knowledge, official guidelines regarding the use of vaginal seeding are not available in other countries to date. However, trials outside the USA including NCT0392843 in Stockholm, Sweden, and a trial in New Zealand, follow similar screening criteria for infectious agents, as trials conducted within the USA [106].”

Reviewer 3 Report

Comments and Suggestions for Authors

In this review by LaPoint et al, the authors summarized evidence from observational studies and clinical trials on microbiome restoration after C-section and examined emerging research on vaginal seeding, including its effects on immunity, metabolism, neurodevelopment, and underlying mechanisms. The manuscript is well written and thorough. While there are some minor issues (as mentioned below), the manuscript (once revised) should be of interest to the researchers working in this domain.

  1. Line 77 – Please specify whether the age refers to 3 months or 3 years for clarity.
  2. Section Titles – Sections 6 and 7 currently have identical titles; they should be revised to better reflect their distinct content.

Author Response

Response to Reviewer 3 Comments

1. Summary

2. Point-by-point response to Comments and Suggestions for Authors

Comments 1: In this review by LaPoint et al, the authors summarized evidence from observational studies and clinical trials on microbiome restoration after C-section and examined emerging research on vaginal seeding, including its effects on immunity, metabolism, neurodevelopment, and underlying mechanisms. The manuscript is well written and thorough. While there are some minor issues (as mentioned below), the manuscript (once revised) should be of interest to the researchers working in this domain.

  1. Line 77 – Please specify whether the age refers to 3 months or 3 years for clarity.
  2. Section Titles – Sections 6 and 7 currently have identical titles; they should be revised to better reflect their distinct content.

Response 1: Thank you for this positive review.

Point 1: Thank you. For clarity we have changed this to “3 years of age”

Point 2: Thank you for picking this up. Section 7 (now section 9) has been changed to “Evidence of the effect of vaginal seeding on infant health outcomes”

Reviewer 4 Report

Comments and Suggestions for Authors

Dear Authors,

The Manuscript „Can vaginal seeding at birth improve health outcomes of Cesarean section delivered infants? A scoping review” is well written in accordance with the guidelines of the journal. The title is accurate and relevant. The paper has an informative abstract. All figures and tables are necessary and understandable. The names of organisms are used appropriately. The data were systematically analyzed. The discussion and conclusions are correct and reflect the evidence provided in the paper with a few necessary additions. The references were accurate. It would be good to add the following information to improve the quality of the paper:

  1. The difference in the gut microbiome of the two groups of newborns after normal vaginal birth and after Cesarean section, when they are breastfed and when they are formula fed, has not been sufficiently explained. It would be good to provide additional information about the change in the microbiome with the different type of feeding and how it affects children born by Cesarean section, whether convergence with the other group occurs. More data are needed for comparison and whether a statistical difference is established between the two groups with both types of diets.
  2. Interesting are the data on a more frequent association of certain diseases such as obesity, Insulin resistance, diabetes, asthma, allergic reactions, autism, and even a higher incidence of certain oncological and hematological diseases with the microbiome of children born by Cesarean section. But are there other predisposing factors that are not discussed and mentioned? These data can be indicated, but they should also be subject to some doubt, without being absolute, and from the discussion it seems that the authors are convinced of this, without subjecting it to further discussion. Additional information should be considered and added to the discussion.
  3. The use of samples from the mother's vaginal or intestinal microbiome to inseminate the newborn, especially together with breast milk, carries a huge risk of infection. This is mentioned very briefly in the article and is ignored because the mother has been tested for the most common sexually transmitted infections. This is not enough at all, because there are microorganisms such as some viruses, Mycobacterium spp., Listeria spp., E coli K1, which are not routinely looked for and others which can colonize the vagina or gut in small quantities and may be missing from the result. Some of them are important caustive agents of neonatal meningitidis such as Listeria spp., E coli K1, Streptococcus agalactiae. Herpes viruses, Coxsacie viruses and many others can cause encephalitis with high mortality in newborns and sometime autoimmunity reactions, and clinical type 1 diabetes. Even weakly virulent bacteria can cause bacteremia in newborns, due to the high permeability of the intestinal mucosa, and there have been cases described even from an overdose of probiotics. This method, which can be endanger the life of the newborn, cannot be recommended calmly. The authors must analyze both the benefits and the risks of its application. 

I recommend publishing it in Journal “Microorganisms” with major revision.

18.05.2025.

Author Response

Response to Reviewer 4 Comments

1. Summary

2. Point-by-point response to Comments and Suggestions for Authors

Comments 1: The difference in the gut microbiome of the two groups of newborns after normal vaginal birth and after Cesarean section, when they are breastfed and when they are formula fed, has not been sufficiently explained. It would be good to provide additional information about the change in the microbiome with the different type of feeding and how it affects children born by Cesarean section, whether convergence with the other group occurs. More data are needed for comparison and whether a statistical difference is established between the two groups with both types of diets.

Response 1: Thank you for this important point. While we have mentioned the importance of breastfeeding in establishment of the infant gut microbiome in a few places throughout the manuscript, we have now expanded it in the following places to address your point:

Section 3, paragraph 4: “Moreover, breast feeding is known to impact the infant microbiome and have a positive influence on offspring health [6,7,57,58]. Breast-fed infants have slower maturation of the gut microbiome that formula fed infants with increased levels of taxa that are sometimes used as probiotics such as L. johnsonii/L.gasseri, L. paracasei/L. casei, and B. longum [6,7]. Importantly, breast-feeding can mitigate some of the C-section as-sociated disruption to gut microbiome development [58,59]. When receiving breast milk, differences between the gut microbiome between delivery modes became less compared to when receiving formula [58,59]. Recently, “auxiliary” seeding pathways have been described where more beneficial breast milk microbiota can colonize the guts of C-section delivered infants compared to vaginally delivered infants [3,60]. This may be a partial compensatory mechanism to restore the disrupted microbiota following a C-section delivery.”

Section 10: “Safety and Challenges”, paragraph 4: “Additionally, postnatal factors such as delayed initiation of breastfeeding, which is common with C-sections, can alter microbiome development and possibly impact the risk of C-section associated diseases [97,118]. Moreover, breastfeeding itself can positively impact the infant gut microbiome and diminish microbiome differences seen by delivery mode [6,7,58,59]. Robust, well powered trials with randomization so these factors can be balanced in each trial arm are needed to account for this.”

Comments 2: Interesting are the data on a more frequent association of certain diseases such as obesity, Insulin resistance, diabetes, asthma, allergic reactions, autism, and even a higher incidence of certain oncological and hematological diseases with the microbiome of children born by Cesarean section. But are there other predisposing factors that are not discussed and mentioned? These data can be indicated, but they should also be subject to some doubt, without being absolute, and from the discussion it seems that the authors are convinced of this, without subjecting it to further discussion. Additional information should be considered and added to the discussion.

Response 2: Thank you, we agree. We have expanded our discussion to include

Section 4.1: “Numerous studies, systematic reviews, and meta-analyses have reported an association between C-section delivery, higher BMI and risk of developing obesity in both childhood and adulthood, after adjusting for obesogenic risk factors [66-70].”

End of section 4:

“Despite epidemiological evidence showing an association with C-section delivery and an increased risk of some metabolic, immune and neurological disorders, it is also possible that certain factors inherently associated with C-section delivery may predispose to these conditions. For example, delayed initiation of breastfeeding commonly occurs with C-sections and can possibly impact the risk of C-section associated diseases [97]. Additionally, C-section delivery is more common in pregnant women with obesity, which in itself is a risk factor for offspring obesity [98,99]. When evaluating these epidemiological studies, it should be assessed whether possible confounding factors were accounted for.”

Comments 3: The use of samples from the mother's vaginal or intestinal microbiome to inseminate the newborn, especially together with breast milk, carries a huge risk of infection. This is mentioned very briefly in the article and is ignored because the mother has been tested for the most common sexually transmitted infections. This is not enough at all, because there are microorganisms such as some viruses, Mycobacterium spp., Listeria spp., E coli K1, which are not routinely looked for and others which can colonize the vagina or gut in small quantities and may be missing from the result. Some of them are important causative agents of neonatal meningitidis such as Listeria spp., E coli K1, Streptococcus agalactiae. Herpes viruses, Coxsacie viruses and many others can cause encephalitis with high mortality in newborns and sometime autoimmunity reactions, and clinical type 1 diabetes. Even weakly virulent bacteria can cause bacteremia in newborns, due to the high permeability of the intestinal mucosa, and there have been cases described even from an overdose of probiotics. This method, which can be endanger the life of the newborn, cannot be recommended calmly. The authors must analyze both the benefits and the risks of its application. 

Response 3: Thank you for this important point. While we do think the risk of infection from vaginal flora is low, given the current screening techniques and that children born vaginally are exposed to the vaginal microbiome, we acknowledge these potential risks should be expanded on.

Section 10 “Safety and Challenges”, paragraph 2 has been expanded to read:

“For example, approximately 60%–80% of women who deliver a Herpes Simplex Virus (HSV)-infected infant have no evidence of genital HSV infection at time of birth, along with no genital herpes history among themselves or their partner [116]. In addition, the vaginal microbiome can harbor other potential pathogens which are not tested for in clinical trials, including Escherichia coli containing the K1 antigen, which may cause invasive disease in vulnerable newborns [117]. Furthermore, fecal transplant studies have shown that even healthy individuals can harbor low levels of pathogenic organisms which can become disease causing when transferred to a vulnerable individual, thus resulting in increased infectious screening required for fecal transplant [118].”

Section 10 “Safety and Challenges”, paragraph 3: “Nevertheless, this emphasizes that for now vaginal seeding should only be performed under approved clinical trials where infections are carefully screened for and safety data followed. Careful attention should be given to infectious diseases that are ac-quired that are not in the current screening protocols, to assess if they were transmit-ted via vaginal seeding and should be added to screening protocols. This also emphasizes the need to store study materials such as the vaginal gauze used for seeding to be able to access for possible causation.”

Round 2

Reviewer 2 Report

Comments and Suggestions for Authors

Good manuscript

Reviewer 4 Report

Comments and Suggestions for Authors

Dear authors,

Thank you for the corrections and additions made. Now the information is more comprehensive, the conclusions sound more competent.